# High Frequency Hysteresis Losses on γ-Fe_2_O_3_ and Fe_3_O_4_: Susceptibility as a Magnetic Stamp for Chain Formation

**DOI:** 10.3390/nano8120970

**Published:** 2018-11-24

**Authors:** Irene Morales, Rocio Costo, Nicolas Mille, Gustavo B. da Silva, Julian Carrey, Antonio Hernando, Patricia de la Presa

**Affiliations:** 1Instituto de Magnetismo Aplicado (UCM-ADIF-CSIC), P.O. Box 155, Las Rozas, 28230 Madrid, Spain; irenemorales@ucm.es (I.M.); antonio.hernando@externos.adif.es (A.H.); 2Instituto de Ciencia de Materiales de Madrid/CSIC, Sor Juana Inés de la Cruz 3, 28049 Madrid, Spain; rocio.costo@gmail.com (R.C.); gusbezerra@ufrrj.br (G.B.d.S.); 3Université de Toulouse, INSA, UPS, Laboratoire de Physique et Chimie des Nano-Objets (LPCNO), 135 Avenue de Rangueil, F-31077 Toulouse, France and CNRS, UMR 5215, LPCNO, F-31077 Toulouse, France; mille@insa-toulouse.fr (N.M.); julian.carrey@insa-toulouse.fr (J.C.); 4Departamento de Física de Materiales, Universidad Complutense de Madrid, 28048 Madrid, Spain

**Keywords:** hyperthermia, magnetic nanoparticles, specific absorption rate, iron oxide

## Abstract

In order to understand the properties involved in the heating performance of magnetic nanoparticles during hyperthermia treatments, a systematic study of different γ-Fe_2_O_3_ and Fe_3_O_4_ nanoparticles has been done. High-frequency hysteresis loops at 50 kHz carried out on particles with sizes ranging from 6 to 350 nm show susceptibility χ increases from 9 to 40 for large particles and it is almost field independent for the smaller ones. This suggests that the applied field induces chain ordering in large particles but not in the smaller ones due to the competition between thermal and dipolar energy. The specific absorption rate (SAR) calculated from hysteresis losses at 60 mT and 50 kHz ranges from 30 to 360 W/g_Fe_, depending on particle size, and the highest values correspond to particles ordered in chains. This enhanced heating efficiency is not a consequence of the intrinsic properties like saturation magnetization or anisotropy field but to the spatial arrangement of the particles.

## 1. Introduction

Magnetic fluid hyperthermia has been intensively investigated since, in 1993, Jordan et al. [1] reported on the potential applications of colloidal monodomain ferrite particle suspensions (“magnetic-fluid”) subjected to an alternating magnetic field as nanoheaters for hyperthermia cancer treatment. Thenceforth, the investigation on particle composition, size and shape, size distribution [2,3,4,5], physicochemical properties [6,7,8,9], media viscosity [10,11,12,13], magnetic properties [14,15,16], toxicity and biocompatility [17,18], field amplitude and frequency [19], in-vivo and in-vitro experiments [20,21,22], and human trials [23,24] have grown exponentially looking to optimize this new thermal treatment.

The heating efficiencies of nanomaterials are mainly and mostly investigated using calorimetry, i.e., the magnetic fluid is subject to an ac-field and its temperature increase is measured as a function of time [23]. Under adiabatic conditions, the temperature increase of the colloid is proportional to the power loss of the nanoparticles, which in turn is related to their magnetic properties. However, adiabatic setups are unusual [25] and most of the experiments are performed on home-made or commercial setups, which give rise to many uncertainties [26,27] with the consequence of less reproducible experimental results due to the differences in thermal isolation of the colloids. Furthermore, calorimetric experiments give information regarding the influence of magnetic properties on the heating performance but only in an indirect way.

Few years ago, homemade ac-magnetometers have been developed in order to measure hysteresis loops under radiofrequency fields [28,29]. This new technology allows for deeper investigation in the understanding of the magnetic properties that optimizes the nanoparticle heating efficiencies and, due to the short measurement times (few seconds), allows a magnetic characterization at almost constant temperature, giving more reliable results than calorimetry characterizations, in which magnetic properties change with a temperature increase. For example, with this technique, it is possible to study the effect of size, effective anisotropy, colloidal viscosity, or dipolar interactions on the hysteresis cycles [12,30,31,32,33,34,35]. Additionally, by means of numerical simulations, it is possible to investigate different physical conditions that lead to hysteresis losses improvement and compare simulation results with high-frequency hysteresis loops [31,36].

In this work, the heating efficiency of maghemite (γ-Fe_2_O_3_) and magnetite (Fe_3_O_4_) with sizes ranging from 6 to 350 nm are investigated under ac- and dc-magnetometry. The hysteresis loops under high-frequency field for particle of 35 nm in size show an increase of the volume susceptibility, χ, with applied field that can be associated to chain formation, making these particles the most efficient nanoheaters. On the contrary, for smaller particles, thermal energy inhibits chain formation.

## 2. Materials and Methods

### 2.1. Synthesis of the Iron Oxide Nanoparticles

Magnetite particles of 35 nm (Fe_3_O_4_-35nm) are synthesized by oxidative precipitation of FeSO_4_ in the presence of ethanol [37]. The FeSO_4_ precipitation and the subsequent aging are carried out in a globe box to avoid the formation of other undesirable secondary iron phases due to an oxidizing atmosphere. To synthesize the Fe_3_O_4_-35nm sample, two solutions were prepared: a basic solution consisting on a mixture of 25 mL NaNO_3_ (2 M) and NaOH (4.2 M), 88 mL of distilled water and 62.5 mL of ethanol, and an acid solution formed by 13.9 g of FeSO_4_·7H_2_O dissolved in 50 mL of H_2_SO_4_ (0.01 M). N_2_ gas was previously passed through all the solutions to ensure that only Fe_3_O_4_ was present in the final precipitate. The basic solution was rapidly added to the acid solution and stirred for 15 min, forming a turquoise-colored compound known as green rust. When the precipitation was completed, green rust was introduced in a jacketed glass bottle previously heated to 90 °C, and the system was closed and undisturbed at this temperature for 24 h. Aging time was fixed at 24 h in order to reach conditions near equilibrium. At this point, the sample was cooled down at room temperature and the solid was separated by magnetic decantation and washed several times with distilled water.

To synthesize magnetite nanoparticles of 14 nm in diameter (Fe_3_O_4_-14nm), 75 mL of a NH_4_OH solution (25%) was rapidly added to a solution of FeCl_2_ (0.175 M) and FeCl_3_ (0.334 M). The addition was carried out rapidly at room temperature under vigorous stirring. Later, the sample was washed three times with distilled water using magnetic decantation.

To obtain the largest nanoparticles (Fe_3_O_4_-350nm), additional time was added as much as possible. Thus, urea (CO(NH_2_)_2_) was used as a base instead of NH_4_OH. The slow hydrolytic degradation of urea in acidic conditions generated ammonia (NH_3_), which increased the reaction pH very slowly, leading to the precipitation of large Fe_3_O_4_ nanoparticles. The reaction vessel was a 100 mL Duran^®^ glass laboratory bottle, which can be used as a low-temperature hydrothermal reactor. An amount of 4.8 g of urea was added to 70 mL of distilled water and stirred vigorously. Then, 2.7 g de FeCl_3_·6H_2_O was added and, once it was totally dissolved, 1.06 g of FeCl_2_·4H_2_O was added to the mixture. To synthesize this sample, N_2_ was bubbled through the solution during the precursor’s addition to reduce the presence of oxygen as much as possible, which produces the oxidation of the iron intermediates and induces the formation of other iron phases different from Fe_3_O_4_. Then, the bottle was closed and introduced in a pre-heated oil bath at 90 °C for 48 h. Sample was under magnetic stirring during the whole process. After this time, the sample was cooled down to room temperature and the resulting black precipitate consisting of Fe_3_O_4_ particles was decanted using magnetic settling and washed several times until the supernatant was totally clear and transparent.

The other types of particles have been synthesized using coprecipitation of a mixture of the Fe(II) and Fe(III) salts in alkaline medium as previously described by other authors in detail [38]. The size of the particles can be controlled by the addition rate and order and also by the aging time and temperature (among others).

In order to enhance the colloidal properties of the particles and to oxidize Fe_3_O_4_ nanoparticles to γ-Fe_2_O_3_, which is more stable at room temperature and atmosphere, an acid treatment was carried out [39]. This treatment consisted of three steps: First, the Fe_3_O_4_ nanoparticles previously synthesized were mixed with 300 mL of HNO_3_ (2 M) and stirred for 15 min. In a second step, the supernatant was removed and Fe(NO_3_)_3_ 1 M (75 mL) and distilled water (130 mL) were added. The mixture was stirred and heated up to the boiling point for 30 min. Then, it was cooled down to room temperature. Finally, in a third step, the supernatant was removed and another 300 mL of HNO_3_ (2 M) were added. The mixture was stirred for 15 min, the supernatant was removed and particles were washed three times with acetone and redispersed in distilled water. Acetone wastes were removed with a rotary evaporator. The maghemite nanoparticles obtained as a result of the oxidation of magnetite were named as γFe_2_O_3_-8nm. Longer stirring times during the third step led to the smallest nanoparticles (sample γFe_2_O_3_-6nm) as the particle surface was partially dissolved by nitric acid.

Slight modifications of this synthesis protocol led to larger nanoparticles. Thus, to synthesize the γFe_2_O_3_-12nm sample, Fe(II)-Fe(III) solution was added to the basic solution as slow as possible (drop by drop). In addition, the aging time was increased from 5 min to 1 h and the aging temperature was fixed to 90 °C. Afterwards, the oxidizing acid treatment was carried out as explained previously [10].

### 2.2. Structural and Colloidal Characterization

The crystal structure of the samples was identified by X-ray diffraction (XRD) performed in a Bruker D8 Advance diffractometer with a graphite monochromator using CuKα radiation (λ = 1.5406 Å). The patterns were collected within 10° and 90° in 2θ. The XRD spectra were indexed to an inverse spinel structure. The average particle size is calculated using Scherrer´s formula using the half width of the (311) X-ray diffraction line, except for sample Fe_3_O_4_-350nm, whose large size prevented the use of Scherrer´s formula for size determination.

Particles size and shape were studied by transmission electron microscopy (TEM) using two different apparatuses: a JEM-200 FX microscope operated at 200 keV and a JEM1010 microscope operated at 100 kV (both from Japan Electron Optics Laboratory Company Limited; Tokyo, Japan). TEM samples were prepared by placing one drop of a dilute particle suspension on an amorphous carbon-coated copper grid and evaporating the solvent at room temperature. The mean particle size of each sample was calculated by measuring the largest internal dimension of at least 100 particles. Afterward, data were fitted to a log normal distribution by obtaining the mean size *d* and standard deviation *σ*.

Infrared spectra of the samples diluted in KBr at 1% were recorded between 3600 and 300 cm^−1^ in a IFS 66V-S from Bruker (Massachusetts, United States). Iron concentration determination of all the samples is carried out on an inductively coupled plasma—optical emission spectrometer (ICP-OES) model OPTIME 2100DV from Perkin Elmer, Massachusetts, United States.

Colloidal properties of the samples were studied in a Zetasizer Nano S, from Malvern Instruments (Malvern, Worcestershire, United Kingdom). The hydrodynamic size of the particles in suspension was measured using dynamic light scattering (DLS) in intensity (Z-average) in acidic medium. Each hydrodynamic value was the result of three different measurements at different dilutions to avoid errors coming from back scattering and using the scattering index of water.

### 2.3. Magnetic Characterization

The dc-magnetic characterization was performed in a MPMS-5S SQUID magnetometer from Quantum Design, San Diego, California, United States. The measurements were carried out in special sample holders with 50 μL of the colloidal suspensions. For those samples with a blocking temperature close to room temperature, 5 μL of samples were dropped in a cotton to let the liquid to evaporate. Magnetization curves at 5 T and 10 K, 250 K, or 300 K are measured, as well as zero field-cooled and field-cooled (ZFC-FC) curves with an applied field of 10 mT from 10 to 250 K, in order to keep the sample frozen during the magnetic characterization, or up to 350 K in case of samples in cotton.

The ac-high frequency magnetization curves were measured in a home-made hysteresis loop meter, with a frequency of 50 kHz and magnetic field amplitude ranging between 10 and 60 mT [29].

## 3. Results and Discussion

### 3.1. Structural and Colloidal Characterization

Morphological characteristics of the samples are shown in the TEM micrographs of Figure 1, where the insets show the particle sizes and distributions. As can be seen from Table 1, the size distribution was below the polydispersity degree (standard deviation/mean size) of 0.2 for all particles except the largest ones. Smaller particles (γFe_2_O_3_-6nm, γFe_2_O_3_-8nm, γFe_2_O_3_-12nm, and γFe_2_O_3_-14nm) had a rounded shape, whereas the largest particles (Fe_3_O_4_-35nm and Fe_3_O_4_-350nm) showed rhombohedric or cubic profiles. In light of these micrographs, it seems that particles were not isolated but formed aggregates. In addition, the large particles (Fe_3_O_4_-35nm and specially Fe_3_O_4_-350nm) tended to form chains, as observed in the micrographs (see more TEM images in Appendix A). The particle size observed using TEM was in good agreement with the average crystallite size calculated from the (311) reflection in the XRD pattern by means of Scherrer´s formula, suggesting that the particles were monocrystalline (Table 1) with the exception of Fe_3_O_4_-350nm. A progressive decrease of the diffraction peaks width can be observed in the XRD diffractograms as the particle size increases (see Appendix A). XRD patterns can be indexed to a cubic inverse spinel, either Fe_3_O_4_ or γ-Fe_2_O_3_.

The Fe_3_O_4_ and γ-Fe_2_O_3_ phases could not be discriminated using XRD. Magnetization measurements and Fourier transform infrared spectroscopy analysis (FTIR) were used for this purpose. Saturation magnetization values for samples Fe_3_O_4_-35nm and Fe_3_O_4_-350nm were close to the expected bulk value for Fe_3_O_4_. Concerning sample Fe_3_O_4_-14nm, previous studies have shown that particles produced by coprecipitation were usually formed using a large Fe_3_O_4_ core surrounded by an iron oxide shell as a result of the surface oxidation and disorder [37]. In this case, FTIR helped to determine the phases. Appendix A shows in sample Fe_3_O_4_-14nm two broad bands at 580 and 400 cm^−1^, which is typical for Fe_3_O_4_ [40]. In addition, there was a change in the ratio between bands at 540 cm^−1^ (Fe-O tetrahedral coordination) and 450 cm^−1^ (Fe-O octahedral coordination), which was related to a change in the vacancies distribution during an oxidation process [41]. Therefore, sample γFe_2_O_3_-8nm seemed to be more oxidized than sample Fe_3_O_4_-14nm, and the last one was composed of both iron oxides.

DLS characterization of the γ-Fe_2_O_3_ nanoparticles showed an increase of the hydrodynamic size as particle size increased, with a polydispersity degree σ close to 0.2 for all the samples, suggesting that the nanoparticle aggregation degree was relative low (see Table 1 and Appendix A). In the case of Fe_3_O_4_, the 35 nm nanoparticles formed aggregates of a few particles, contrary to what was expected for large particle sizes (see Table 1 and Appendix A). On the contrary, the 14 nm particles formed larger aggregates than the 35 nm nanoparticles, probably due to the synthesis methods. On the other hand, the aggregates size of the 350 nanoparticles was larger than 2 μm with σ ≈ 0.5, as can be seen in Table 1 and Appendix A. In summary, with the exception of Fe_3_O_4_-350nm, all samples showed hydrodynamic sizes below 100 nm with low polydispersity at acidic pH, indicating good colloidal stability.

### 3.2. Magnetic Properties under dc-Field

Thermal dependence of magnetization and hysteresis loops of Fe_3_O_4_ and γ-Fe_2_O_3_ nanoparticles are shown in Figure 2 and Figure 3 and Appendix A. The saturation magnetization *M_s_*, remanence *M_r_*, blocking temperature *T_b_*, and coercive field *H_c_* values at 10 K extracted from the experimental data are shown in Table 1. Considering the Stoner–Wohlfarth model for a system of uniaxial non-interacting random oriented nanoparticles in single-domain regime, the effective magnetic anisotropy *K_eff_* and blocking temperature can be estimated as follows [42]:(1)$$K_{eff}\approx \mu_{0}H_{c}M_{s}\;\mathrm{and}\;T_{B}\approx \frac{K_{eff}V}{25k_{B}}$$

In the case of small particle sizes, thermal fluctuation can be significant for the calculation of the anisotropy field *H_k_* and it has to be considered. The following formula gives the contribution of the thermal effects to *H_c_* [16]:(2)$$\begin{array}{l} \mu_{0}H_{c}=0.48\;\mu_{0}H_{k}(1-\kappa^{0.8})\\ \kappa =\frac{k_{B}T}{K_{eff}V}\ln\left(\frac{k_{B}T}{\mu_{0}H_{\max}M_{s}Vf\tau_{0}}\right)\\ \underset{T\to 0}{\lim}\mu_{0}H_{c}=0.48\;\mu_{0}H_{k} \end{array}$$
where $f\approx 10^{-4}\;\mathrm{Hz}$ is the measuring frequency of the SQUID.

For the smallest particles of 6 and 8 nm, the *M_s_* values in the hysteresis loops (see insets Appendix A) were smaller than the bulk values [43] but close to the expected ones for small particles [10]. For sizes larger than 12 nm, *M_s_* was close to the bulk.

*M_r_* was another relevant property of the hysteresis loops that gives information about the magnetic particles interactions. Following the Stoner–Wohlfarth model [42], $M_{r}/M_{s}=\;0.5$ for a system of non-interacting single domain (SD) particles; thus, any deviation from this ratio gives information about the sort of interactions between particles. If this ratio is smaller than 0.5, dipolar interactions of random oriented nanoparticles can be considered responsible for the decrease; however, if the ratio is larger than 0.5, it is a signal of magnetic coupling as in the case of chain arrangement [31,44,45]. Under dc-field, $M_{r}/M_{s}\approx \;0.3$ for most of the nanoparticles, suggesting that they were randomly oriented and subjected to dipolar interactions. However, for the 6 and 8 nm nanoparticles, $M_{r}/M_{s}<\;0.3$ because there were still unblocked magnetic moments at 10 K due to thermal fluctuations.

From Figure 2, it is observed that, in the case of γ-Fe_2_O_3_, *T_b_* increased with particles size, as expected: 70 K for the 6 nm particles, 90 K for the 8 nm, 220 K for 12 nm, and around room temperature for the 14 nm. However, *T_b_* >> 300 K for Fe_3_O_4_ nanoparticles of 14 and 35 nm. Note that Fe_3_O_4_ and γ-Fe_2_O_3_ of 14 nm have quite different *T_b_* despite having the same size. For temperatures below 50 K, the incipient change in the ZFC slope of Fe_3_O_4_-35nm is characteristic for magnetic nanoparticles with sizes ranging from 20 to 50 nm and associated to the Verwey transition [46,47]. For the largest Fe_3_O_4_-350nm particles, the Verwey transition was observed around 120 K.

Hysteresis loops at room temperature show nearly superparamagnetic behavior with negligible *H_c_* (<1 mT) and *M_r_* for particles smaller than 14 nm (see Appendix A). At low temperatures, *H_c_* increased with particle size in the case of γ-Fe_2_O_3_: 7.5, 11.0, 25.5, and 25.1 mT for the 6, 8, 12, and 14 nm sizes, respectively. In contrast, in the case of Fe_3_O_4_, *H_c_* showed the opposite behavior: it decreased when increasing the particle size probably due to the formation of a flux closure state with small remanence. Figure 4 shows the size dependence of *H_c_*. It is well established that the SD size $d_{SD}<90\;\mathrm{nm}$ for Fe_3_O_4_ and γ-Fe_2_O_3_ [48], and as shown in Figure 4, all samples were SD with the exception of Fe_3_O_4_-350nm.

For the 6 and 8 nm nanoparticles, the decrease of *H_c_* with decreasing particle size was given mainly by thermal fluctuations. As can be seen in Figure 5, the thermal fluctuations were significant for particles smaller than 8 nm, but negligible for particles larger than 12 nm with $\mu_{0}H_{c}\approx 0.48\;H_{K}$, as in the 0 K problem. By means of Equation (2), *H_k_* could be calculated as a function of particle size *d* (see Table 1) or as a function of variable κ that contained information regarding the temperature and the intrinsic magnetic properties.

As can be seen from Figure 2 and Figure 3, for particle sizes close to 14 nm, γ-Fe_2_O_3_ showed superparamagnetic behaviour but Fe_3_O_4_ did not: *T_b_* was close to room temperature for γ-Fe_2_O_3_ but above 350 K in the case of Fe_3_O_4_ (see Figure 3). Even when the sample Fe_3_O_4_-14nm was not subjected to any oxidative process, FTIR results showed a mixture of Fe_3_O_4_ and γ-Fe_2_O_3_ (see Appendix A), which suggested that the as-synthesized sample Fe_3_O_4_ suffered a spontaneous oxidation at the surface, giving rise to a kind of core/shell structure. In the case of nanoparticles, the presence of a shell can induce surface anisotropy that increases *K_eff_* due to the high surface-to-volume ratio [41], which was reflected in a high *H_c_* of this sample. Besides magnetic interactions, this enhanced *K_eff_* could also explain the higher *T_b_* in Fe_3_O_4_-14nm compared to γFe_2_O_3_-14nm.

### 3.3. High-Frequency Hysteresis Loop Measurements

Unlike dc-measurements where the particles are in a solid matrix (either ice or cotton) and the only possible relaxation mechanism is by Neel, the ac-measurements are performed in the colloidal suspension; therefore, both Neel and Brown relaxations can be present. It has been previously reported that the relaxation mechanism of γ-Fe_2_O_3_ synthesized using a co-precipitation method is by Neel for particles with a size below 12 nm, whereas, due to size distribution, Neel and Brown relaxations are present for particles larger than 12 nm [10]. In the case of Fe_3_O_4_, the large magnetic anisotropy of the 14 nm nanoparticles allows one to assume that the main relaxation mechanism is Brownian, as well as that of the 35 nm nanoparticles, but in this case it was due to the size.

High-frequency hysteresis loops were measured at different applied fields in order to determine the dependence of hysteresis losses with field amplitude. The γ-Fe_2_O_3_-6nm and multidomain Fe_3_O_4_-350nm were excluded from this analysis because it was not possible to obtain reproducible hysteresis loops due to the low magnetization of the samples under an ac-field. Figure 6 and Figure 7 show the hysteresis curves for the γ-Fe_2_O_3_ and Fe_3_O_4_ samples, respectively. As can be seen, the areas under the curves increased with increasing applied field, with the exception of γFe_2_O_3_-8nm, which displayed an almost perfectly reversible hysteresis loop, due to the low *T_b_* of this sample (*T_b_* ≈ 90 K) (see Figure 6A). According to the linear response theory, the maximum area of the hysteresis loops took place when ω*τ_R_* ≈ 1 [16]. By calculating the relaxation time *τ_R_* of the 8 nm maghemite nanoparticles as $\tau_{R}=\tau_{0}e^{\frac{K_{eff}V}{k_{B}T}}$ with *τ*_0_ ≈ 10^−9^–10^−11^ and *K_eff_* calculated using Equation (1), it is found that ω*τ_R_* << 1. Therefore, the sample had a fast relaxation even at high frequency, so the ac-hysteresis cycles still showed near superparamagnetic behavior at this frequency.

Samples γFe_2_O_3_-12nm and γFe_2_O_3_-14nm showed non-saturated ac-hysteresis cycles (Figure 6B,C), even for $\mu_{0}H_{\max}>\mu_{0}H_{k}$. The coercive field values at maximum ac-field were 5.5 and 10.6 mT for the γFe_2_O_3_-12nm and γFe_2_O_3_-14nm, respectively, indicating that there were more particles blocked in γFe_2_O_3_-14nm. Considering that the samples had similar $\mu_{0}H_{k}$ under a dc-field (see Table 1) but *T_b_* was smaller for γFe_2_O_3_-12nm than γFe_2_O_3_-14nm, it is clear that, due to size distribution, there existed a higher superparamagnetic contribution in γFe_2_O_3_-12nm.

High-frequency hysteresis loops for magnetite samples are shown in Figure 7. At low fields, Fe_3_O_4_-14nm showed an almost linear dependence of magnetization with magnetic field, the loops began to open for moderate fields, and finally, non-saturated hysteresis loops were observed at high fields. There were two possible reasons for the observation of non-saturated hysteresis loop: (i) a superparamagnetic behavior of the magnetization, because a Langevin function could not saturate even if $\mu_{0}H_{\max}>\;\mu_{0}H_{k}$, as in the previous cases, or (ii) $\mu_{0}H_{\max}<\;\mu_{0}H_{k}$, i.e., a much higher ac-field was needed for reaching the saturation.

Under a dc-field, this sample showed *T_b_* > 350 K and $\mu_{0}H_{c}\left(10\;K\right)=39\;\mathrm{mT}\;\left(\mu_{0}H_{k}<\;84\;\mathrm{mT}\right)$; therefore, superparamagnetic behavior could be discarded and the origin of non-saturated hysteresis loops under an ac-field seems to lie in the high *H_k_*, with $\mu_{0}H_{\max}<\;\mu_{0}H_{k}$.

On the contrary, Fe_3_O_4_-35nm showed almost ellipsoidal hysteresis loops for low fields, the areas became higher as applied field increases, and, finally, the saturation magnetization was reached for the maximum applied field $\mu_{0}H_{\max}=60\;\mathrm{mT}$, a value close to the anisotropy field $\mu_{0}H_{k}=57.7\;\mathrm{mT}$. For these large particles, the main relaxation mechanism was Brownian.

It is worth noting that, at room temperature, $M_{s}^{ac}$ for ac-fields were much smaller than $M_{s}^{dc}$ for dc-fields for all samples except Fe_3_O_4_-35nm (see Table 1). For most samples, this was a consequence of the value of the applied field, $\mu_{0}H_{\max}=60\;\mathrm{mT}$, as can be seen in Figure 8A for γFe_2_O_3_-12nm. When $\mu_{0}H_{\max}=60\;\mathrm{mT}$, the samples showed the same magnetization for either ac- or dc-fields. However, for the largest particles, $M_{s}^{ac}=78{\;\mathrm{Am}}^{2}/\mathrm{kg}$ at 60 mT was close to $M_{s}^{dc}=75{\;\mathrm{Am}}^{2}/\mathrm{kg}$ reached when the dc-field was as high as 3 T, suggesting that particle interactions taking place at ac-field could have been responsible for the saturation of the magnetization in lower fields in an ac-field.

In this sample, $M_{r}/M_{s}\approx 0.7$, which is in contrast to the other ones, for which $M_{r}/M_{s}\approx 0.3$ under ac-fields. These enhanced values of *M_s_* and *M_r_*/*M_s_* suggest that, for large particles, a possible chain ordering took place under ac-fields, as it has been shown that chain formation improves the hysteresis losses [31,36,44,45,49]. Moreover, it has recently been reported that a cubic iron oxide nanoparticle arranged in a chain can display an extremely augmented anisotropy due to the collective response of the system [5]. In fact, this augmented anisotropy is not an intrinsic property of the particle but could be given by the demagnetizing factor of the chain.

In order to analyze the evident differences in the magnetic response under an ac-field, the susceptibility χ at *H_c_* was calculated by fitting the slope of the hysteresis loops around *H_c_* for samples γFe_2_O_3_-12nm and Fe_3_O_4_-35nm, as shown in Figure 9. Two very different behaviors were observed for small and large particles: (i) for 12 nm NPs, χ ≈ 9 was independent of the applied field, (ii) in the case of the large ones, χ ≈ 9 for low fields and increased up to χ ≈ 40 in high fields. Additionally, for the large particles, the susceptibility increase had two different rates: below 30 mT, χ increased at a rate of 1.5 mT^−1^ and above 30 mT at 0.2 mT^−1^. Chain formation changed the demagnetizing field and induced an easy axis in the direction of the field producing a larger squaring of the hysteresis loops, as observed in Figure 9. Consequently, a possible chain formation could be associated with the increase of the susceptibility in the large particles.

The question is why this effect was only observed in the large particles but not in the smaller ones. The competition between dipolar energy $E_{d}=\frac{\mu_{0}m^{2}}{4\pi r^{3}}$, with *m* and *r* being the average magnetic moment and separation, respectively, and thermal energy $E_{T}=k_{B}T$ could explain the inhibition of chain formation for the smaller particles. Assuming homogeneously distributed nanoparticles, the dipolar energy can be estimated by setting $E_{d}\approx E_{T}$ and $E_{d}\;\gg \;100\;E_{T}$ for the 12 nm and 35 nm NPs, respectively. It seems that, in the case of iron oxides, thermal fluctuations inhibited the formation of chains when the thermal energy was comparable to the dipolar one. If this is the case, the chain formation depended not only on *V*, *M_s_*, and *H*, but also on *T*. Low temperatures would promote the formation of chains, and high temperatures would inhibit them. The two different regimes in the susceptibility increase could be associated to different processes involved in the chain formation: (i) the minimum energy to overcoming the dipolar interactions of initially random oriented nanoparticles, and (ii) the length of the chain, as reported by Serantes at al. [44]: the longer the chain, the higher the susceptibility and remanence are.

### 3.4. Specific Absorption Rate (SAR)

*SAR* can be determined by the area *A* of the high-frequency hysteresis loops [16] as SAR=A·f, where *f* = 50 kHz for these experiments. Figure 10 shows *SAR* and *A* values for samples γFe_2_O_3_-8nm, γFe_2_O_3_-12nm, γFe_2_O_3_-14nm, and Fe_3_O_4_-35nm, with both values given per Fe mass in order to compare with other reported results. It is worth noting that the differences in heating efficiency for the last sample compared to the rest of them. For field values close to the biomedical application range (*μ*_0_*H* < 20 mT), the *SAR* values were below 100 W/g, close to that reported previously for iron oxides synthesized by co-precipitation method [10,50]. For higher fields, the Fe_3_O_4_-35nm was much more efficient than the smaller ones; it is worth noting that this was not related to the intrinsic properties of the particles but to the capability of these particles to form chains and thus increasing the area under the hysteresis loops [51]. As can be seen from Figure 11, *SAR* shows a quadratic dependence with applied field up to $\mu_{0}H=30\;\mathrm{mT}$. For higher fields, as the hysteresis loops reached saturation, there was still a weak linear dependence of the *SAR* with the applied field, which can be related to the susceptibility increase (see Figure 9).

To illustrate the effect of the anisotropy field on the heating efficiency, the *SAR*s and *A* of the samples γFe_2_O_3_-14nm and Fe_3_O_4_-14nm with the same size but different *H_k_* are compared in Figure 12. Even when they have the same size, γFe_2_O_3_-14nm was more efficient in heating because $\mu_{0}H_{\max}\approx \mu_{0}H_{k}$, whereas for Fe_3_O_4_-14nm, $\mu_{0}H_{\max}\approx 0.7\;\mu_{0}H_{k}$, which is far from saturation.

## 4. Conclusions

This work presents results on the heating efficiencies of Fe_3_O_4_ and γ-Fe_2_O_3_ particles produced using a co-precipitation method with sizes ranging from 6 to 350 nm. The results show that particles of 8 nm still showed superparamagnetic behaviour under an ac-field of 60 mT at 50 kHz, showing very low heating efficiency. For 12, 14, and 35 nm particles, the heating efficiencies strongly depended on their magnetic properties. γ-Fe_2_O_3_ of 12 nm with *T_b_* ≈ 220 K show non-saturated hysteresis losses under an ac-field due to a superparamagnetic contribution coming from still unblocked nanoparticles, even for a field as high as 60 mT. Fe_3_O_4_ and γ-Fe_2_O_3_ of the same size (*d* = 14 nm) have a markedly different magnetic response under ac-fields due to the relationship between $\mu_{0}H_{\max}$ and $\mu_{0}H_{k}$: $\mu_{0}H_{\max}\approx \mu_{0}H_{k}$ for the 14 nm γ-Fe_2_O_3_ whereas $\mu_{0}H_{\max}<\mu_{0}H_{k}$ for the 14 nm Fe_3_O_4_, making the former much more efficient than the latter ones.

From all particles studied here, 35 nm magnetite particles were the best nanoheaters. The application of high-frequency fields increased χ from 9 to 40. This effect was attributed to the formation of chains under the influence of the applied magnetic field. This chain formation was not observed in smaller particles due to the competition between dipolar and thermal energy: for small particles, the thermal energy was comparable to the dipolar one, inhibiting the formation of chains, whereas in large particles, the magnetic energy overcame the thermal energy. It is worth noting this enhanced heating efficiency was not only a consequence of the particle’s intrinsic properties (like *M_s_*, *H_k_*, etc.), but it was also given by the kind of particle interactions. Therefore, the increase of magnetic susceptibility in the coercive field could be considered as a magnetic stamp for chain formation.

## Figures and Tables

**Figure 1 nanomaterials-08-00970-f001:**
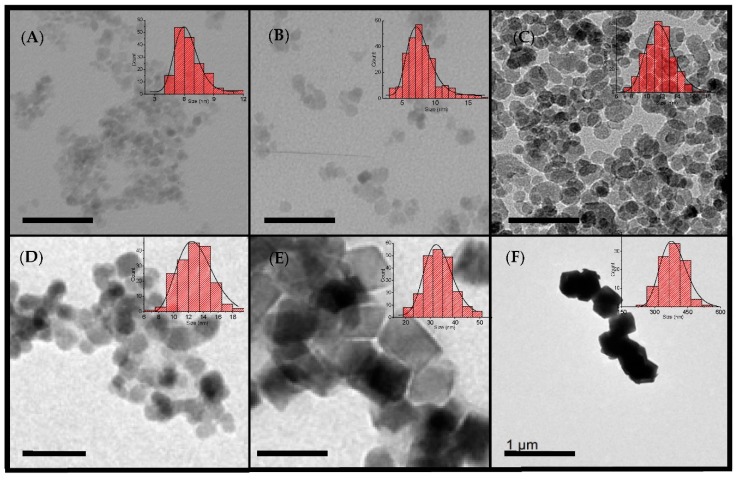
TEM micrographs of γ-Fe_2_O_3_ (**A**) γ-Fe_2_O_3_-6nm, (**B**) γ-Fe_2_O_3_-8nm, and (**C**) γ-Fe_2_O_3_-12nm, and Fe_3_O_4_ particles (**D**) Fe_3_O_4_-14nm, (**E**) Fe_3_O_4_-35nm, and (**F**) Fe_3_O_4_-350nm. Scale bar represents 50 nm except for sample LAU8, where it represents 1 μm.

**Figure 2 nanomaterials-08-00970-f002:**
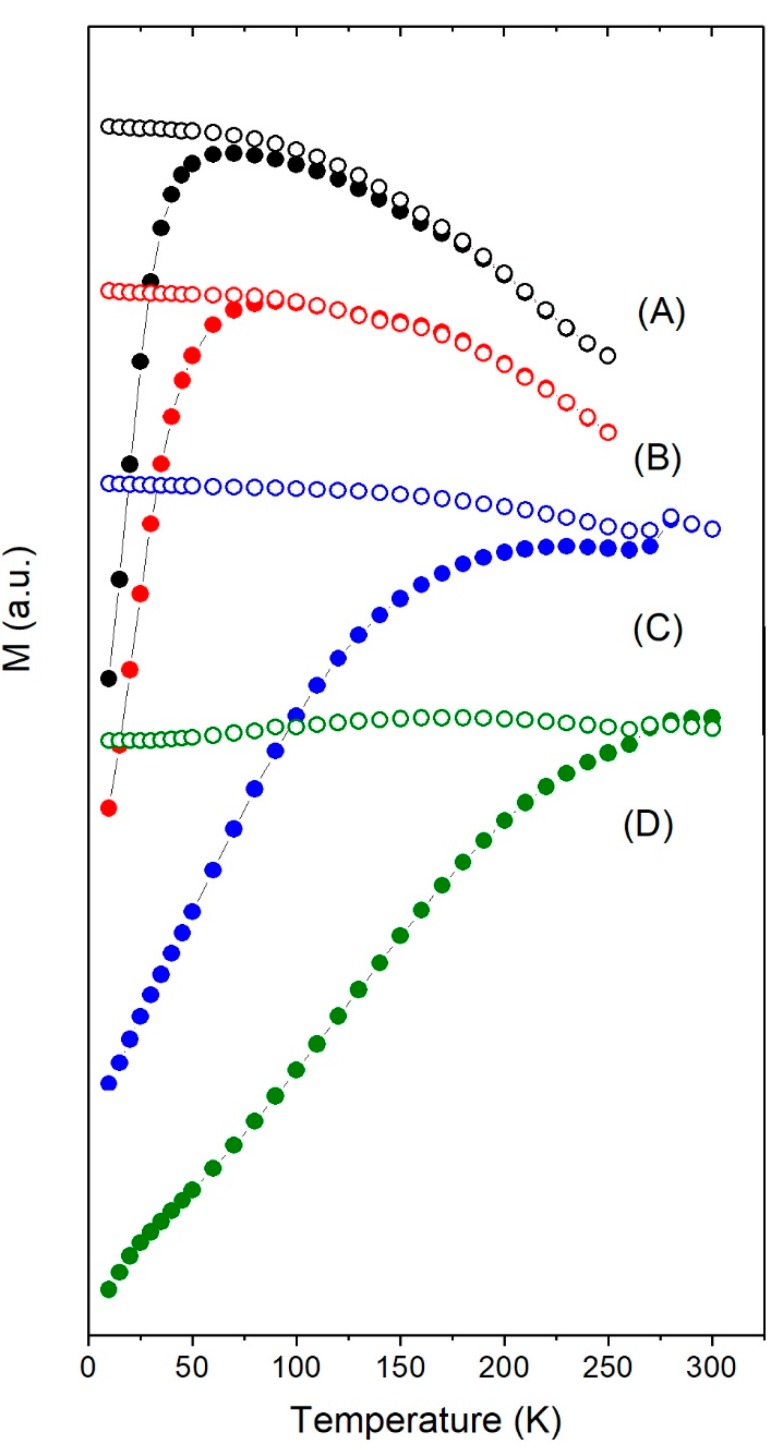
ZFC (fill circles) and FC (open circles) curves under 10 mT applied field for samples (**A**) γFe_2_O_3_-6nm, (**B**) γFe_2_O_3_-8nm, (**C**) γFe_2_O_3_-12nm, and (**D**) γFe_2_O_3_-14nm.

**Figure 3 nanomaterials-08-00970-f003:**
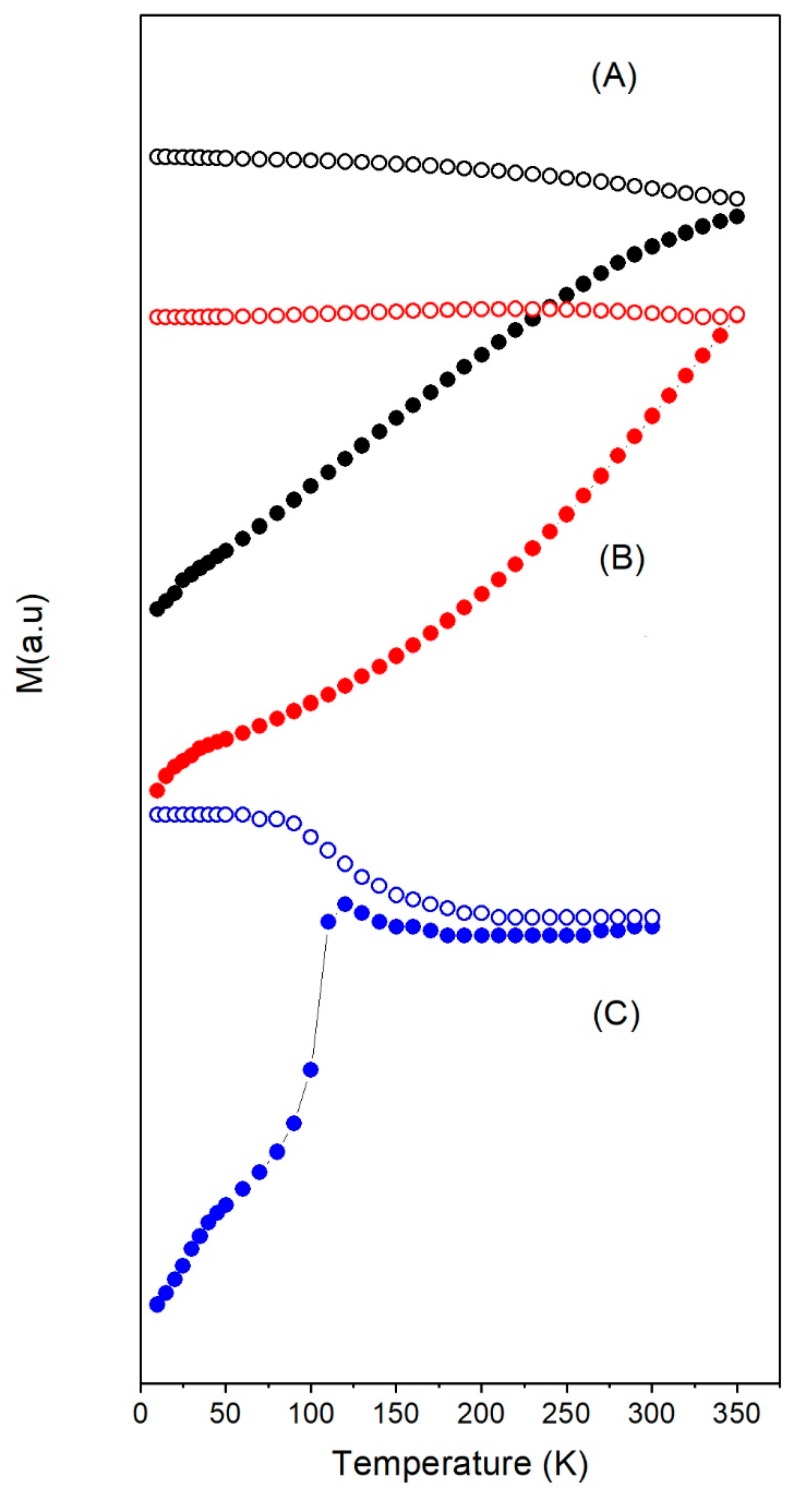
ZFC (filled circles) and FC (open circles) curves under 10 mT applied field for samples (**A**) Fe_3_O_4_-14nm, (**B**) Fe_3_O_4_-35nm, and (**C**) Fe_3_O_4_-350nm.

**Figure 4 nanomaterials-08-00970-f004:**
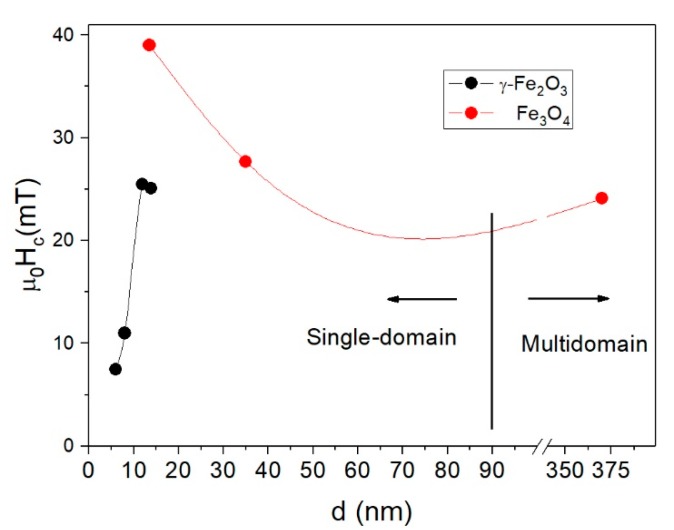
Coercive field vs particle size for all the samples. The vertical line indicates the separation between single-domain and multidomain particles.

**Figure 5 nanomaterials-08-00970-f005:**
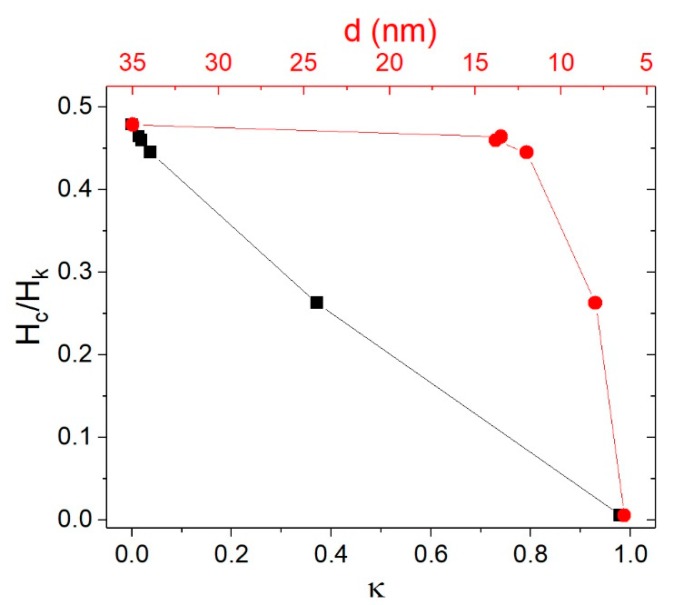
Variation of *H_c_*/*H_k_* with κ (black axis and points) and *d* due to thermal fluctuations (red axis and points) as calculated using Equation (2).

**Figure 6 nanomaterials-08-00970-f006:**
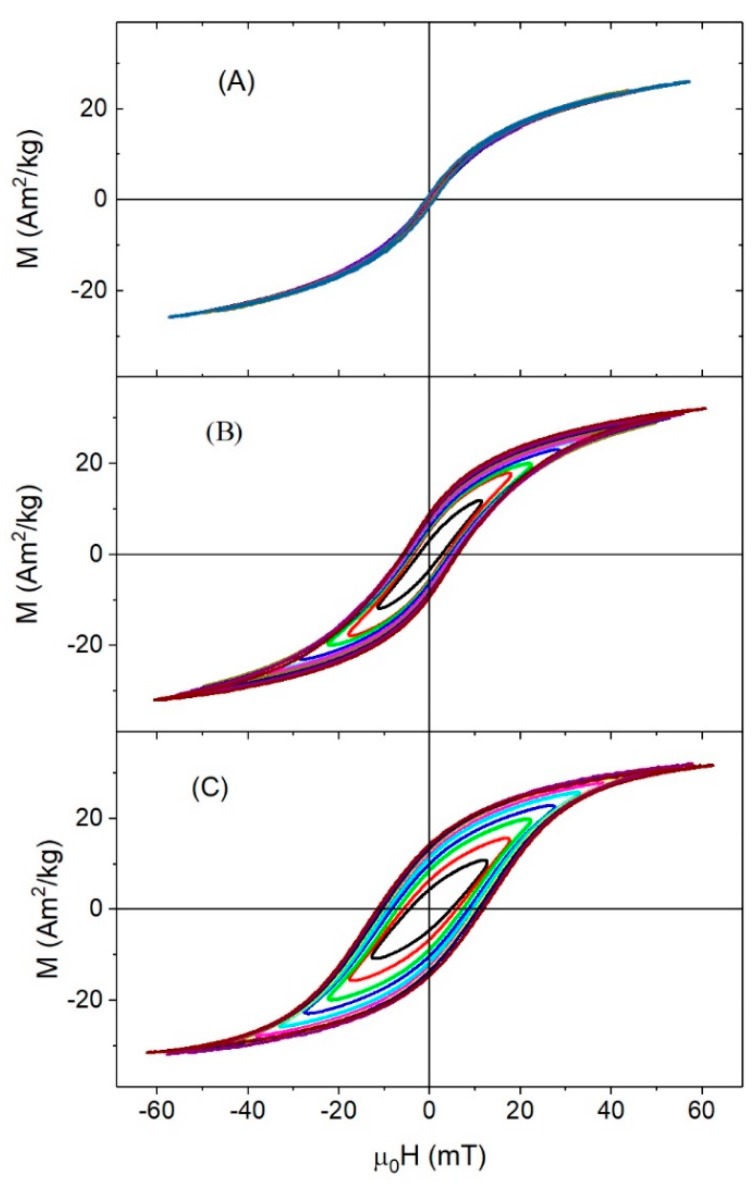
ac-hysteresis loops at room temperatures for γ-Fe_2_O_3_ samples: (**A**) γFe_2_O_3_-8nm, (**B**) γFe_2_O_3_-12nm, and (**C**) γFe_2_O_3_-14nm.

**Figure 7 nanomaterials-08-00970-f007:**
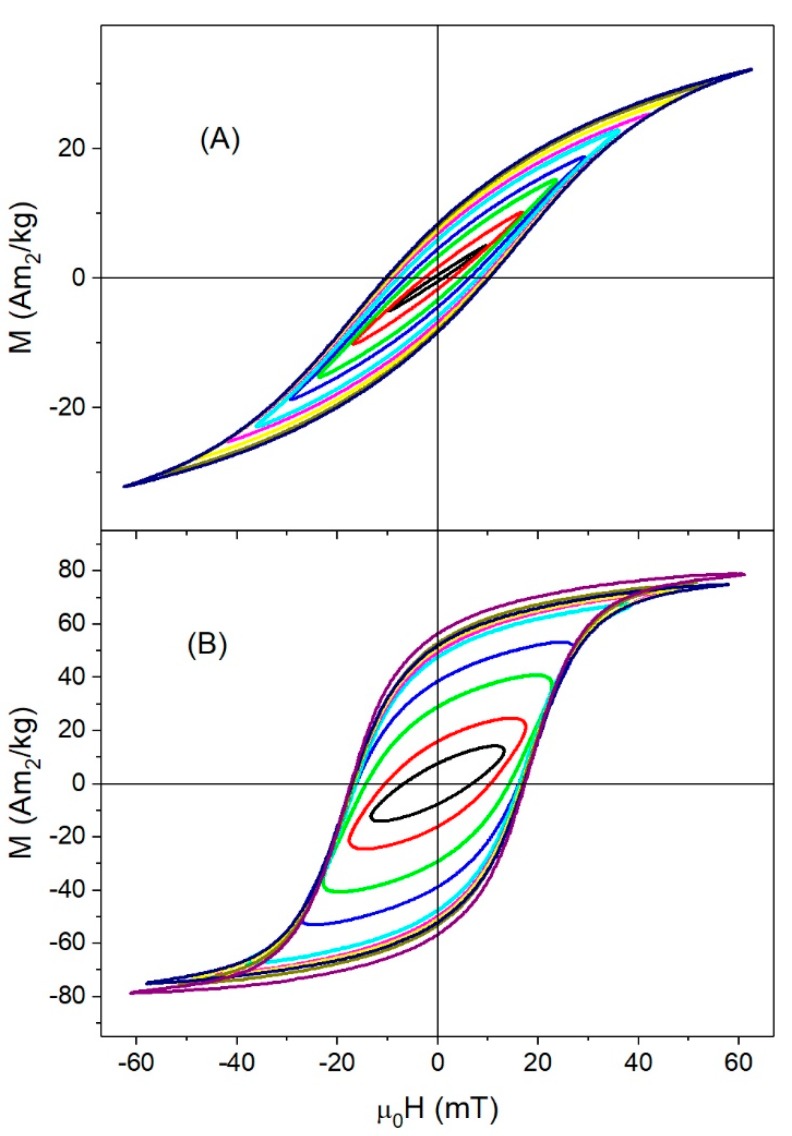
ac-hysteresis loops at room temperatures for Fe_3_O_4_: (**A**) Fe_3_O_4_-14nm, and (**B**) Fe_3_O_4_-35nm.

**Figure 8 nanomaterials-08-00970-f008:**
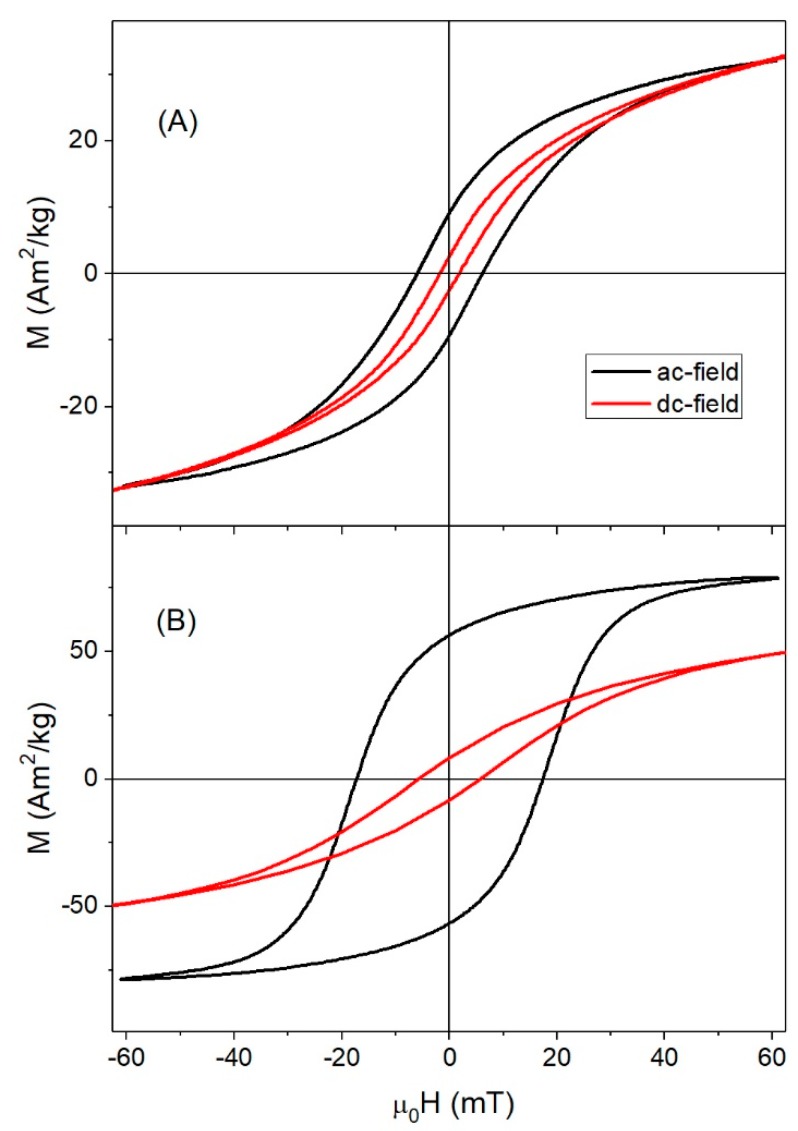
Comparison of magnetization at 60 mT for ac- (black) and dc- (red) measurements in (**A**) γFe_2_O_3_-12nm , and (**B**) Fe_3_O_4_-35nm samples. (Note: The dc-hysteresis curves were measured at 250 K).

**Figure 9 nanomaterials-08-00970-f009:**
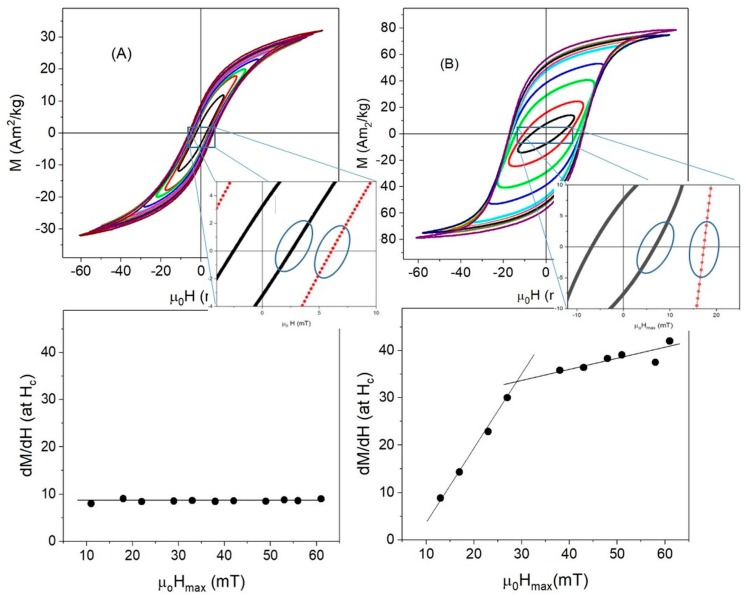
χ calculated at *H_c_* with increasing ac-fields in samples (**A**) γFe_2_O_3_-12nm, and (**B**) Fe_3_O_4_-35nm.

**Figure 10 nanomaterials-08-00970-f010:**
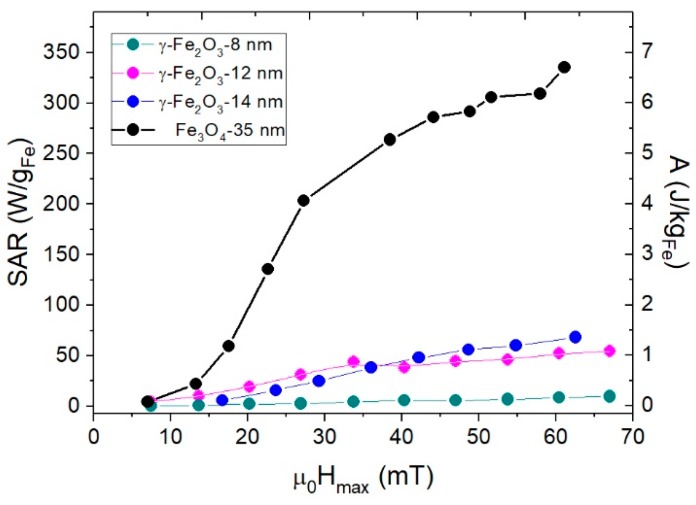
Areas *A* of the hysteresis cycles and *SAR*s calculated from magnetometric measurements for γFe_2_O_3_-8nm, γFe_2_O_3_-12nm, γFe_2_O_3_-14nm, and Fe_3_O_4_-35nm.

**Figure 11 nanomaterials-08-00970-f011:**
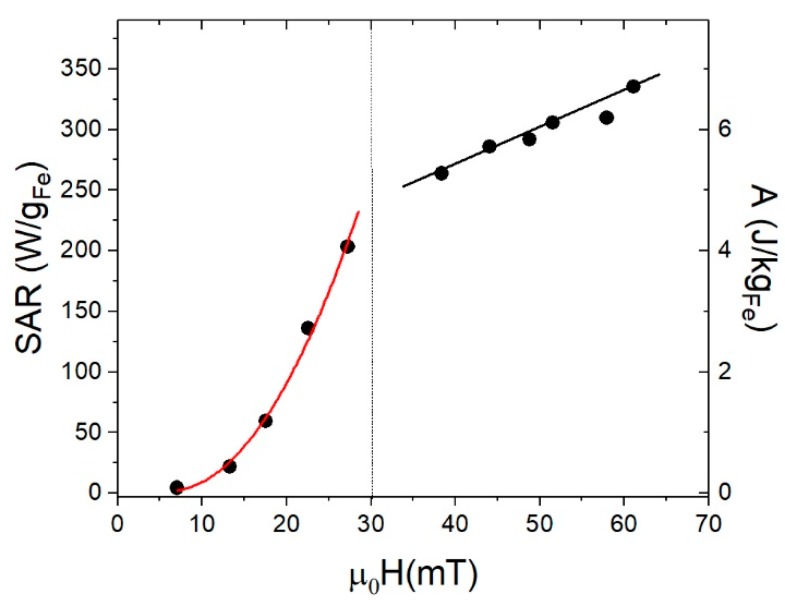
*SAR* (**left**) and *A* (**right**) calculated from ac-magnetometry for Fe_3_O_4_-35nm. The vertical line at 30 mT is a visual guide.

**Figure 12 nanomaterials-08-00970-f012:**
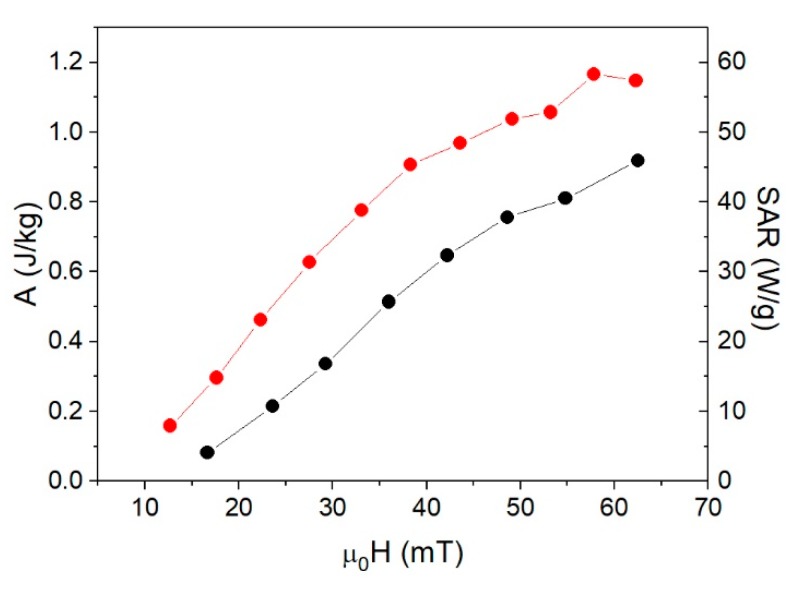
Comparison of areas *A* and *SAR* values calculated as *A·f* for samples γFe_2_O_3_-14nm (red circles) and Fe_3_O_4_-14nm (black circles), both 14 nm in size.

**Table 1 nanomaterials-08-00970-t001:** Size, Structural Phase, and Magnetic Properties of the Nanoparticles ^1^

				*f* ≈ 10^−4^ Hz	*T* = 10 K, *f* ≈ 10^−4^ Hz				*T* = 300 K, *f* = 50 kHz		
Sample Name	TEM Particle Size *d* (nm) (*σ*)	Hydrodynamic Size *D_h_* (nm) (*σ*)	Iron Oxide Majority Phase	*T_B_* (K)	*μ*_0_*H_C_* (mT)	*μ*_0_*H_K_* (mT)	*M_s_* (Am^2^/kg)	*M_r_*/*M_s_*	*μ_0_H_c_* (mT)	*M_S_* (Am^2^/kg)	*M_r_*/*M_s_*
γFe_2_O_3_-6nm	6.3 (0.19)	25.4 (0.27)	γ-Fe_2_O_3_	~70	7.5		55	0.16	—	—	—
γFe_2_O_3_-8nm	7.6 (0.20)	29.9 (0.23)	γ-Fe_2_O_3_	~90	11.0	42	59	0.20	0.9	26	0.05
γFe_2_O_3_-12nm	11.7 (0.16)	58.4 (0.25)	γ-Fe_2_O_3_	~220 K	25.5	57.3	65	0.30	5.5	32	0.28
γFe_2_O_3_-14nm	13.8 (0.18)	96.8 (0.19)	γ-Fe_2_O_3_	~300 K	25.1	54.7	79	0.29	10.6	32	0.45
Fe_3_O_4_-14nm	13.5 (0.19)	160.7 (0.20)	Fe_3_O_4_/γ-Fe_2_O_3_	>300 K	39.0	84.1	71	0.31	10.0	32	0.26
Fe_3_O_4_-35nm	35 (0.20)	88.1 (0.18)	Fe_3_O_4_	>300 K	27.7	57.9	80	0.29	15.8	77	0.72
Fe_3_O_4_-350nm	350 (0.24)	2751 (0.45)	Fe_3_O_4_	>>300 K	24.1		90	0.17	—	—	—

^1^ Particle size *d* and its polydispersity *σ*, hydrodynamic size in Z average *D_h_* and its polydispersity *σ*, iron majority phase, blocking temperature *T_b_*, coercive field *H_c_*, anisotropy field *H_k_*, saturation magnetization *M_s_*, and remanence ratio *M_r_*/*M_s_* of the samples measured at 10 K at low frequency (estimated SQUID measuring time of a hysteresis cycle *t* ≈ 3600 s, therefore *f* ≈ 10^−4^ s) and at 300 K at high frequency (*f* = 52 kHz).

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
