# Peer review of "High Frequency Hysteresis Losses on γ-Fe_2_O_3_ and Fe_3_O_4_: Susceptibility as a Magnetic Stamp for Chain Formation"

_nanomaterials, 2018, doi:10.3390/nano8120970_

Round 1
Reviewer 1 Report
This manuscript investigates the magnetic properties of various species of magnetic nanoparticles, both magnetite and maghemite, synthesized by a combination of methods, so as to cover a wide range of sizes. In addition to a detailed magnetic characterization, the authors also measure the frequency dependent hysteresis loops of the particles, and the SAR values, and provide some evidence of possible formation of chains under the applied magnetic field to explain some otherwise inexplicable results.
The manuscript is rather well written, and worth being considered for publication, as long as the authors address the following comments.
1)The characterization of particles is made by mostly magnetic methods, and electron microscopy. However, these methods are not suitable to determine the colloidal stability of the particles. Therefore, a method to measure the size in solution would be required, such as DLS.
2)When performing an analysis of the frequency dependent magnetic response of magnetic nanoparticles in solution, the authors only consider the Neel relaxation mechanism, and completely ignore the Brownian one, which is usually the dominating one for large particles. Clarifications in this respect would be much appreciated.
3) The method of preparation of the largest particles is very interesting. It is however not clear whether this has been proposed for the first time in this work, or whether it was known form the literature (in which case a citation is required).
Author Response
The replay is uploaded in the word document

Reviewer 2 Report
The work by Presa et.al., reports systematic study of different γ-Fe2O3 and Fe3O4 for magnetic hyperthermia therapy. Authors performed adequate experiments to support their conclusions and can be considered after minor revision.
1. In table 1, what are these values authors has to simplify it its confusing to readers (f » 10-4 Hz, 10 K – f » 10-4 Hz).
2. Figure 1 tem images looks large particle size and high agglomeration.
3. The primary goal of the work is to improve hyperthermic properties of the material, such type of efforts have been done by other authors such as (doi.org/10.1016/j.colsurfb.2013.06.014, 10.1021/acsami.6b16513) by adding different polymers and metals. Authors should refer such reports and reshuffle discussion section.
4. Conclusion paragraph should be modified.
Author Response

(The authors gave the same response as above.)

Reviewer 3 Report
In this manuscript, the authors report the heating efficiencies of Fe3O4 and γ-Fe2O3 particles in the range 6 to 350 nm obtained by co-precipitation method. The results indicate that small particles are superparamagnetic and show very low heating efficiency. For sizes between 12 and 35 nm, the heating efficiencies depend on their magnetic properties. The magnetite particles of 35 nm are the best nanoheaters and this effect is attributed to the formation of chains under the influence of the applied magnetic field (this chain formation is not observed in smaller particles due to competition between dipolar and thermal energy). The work is interesting and I would like to recommend it for publication; however, I suggest considering the following aspects to discuss and imporve the present work:
The different magnetic behaviour is attributed to different parameters of the nanoparticles (size, aggregation, formation of large chains, anisotropy). The characterization of the obtained nanoparticles have been performed to analyse the solid samples using different techniques (FTIR, XRD, TEM); however, the magnetization studies were performed in solution and the changes in magnetization properties were attributed, among other parameters, to the effect of aggregation-disaggregation processes in solution. I do not doubt that magnetization is related to these effects, but experimental measures should be provided to corroborate this. I suggest the study of colloidal stability studies of the different samples are carried out using DLS at the concentrations analysed and corroborate the effects on magnetization properties.
If these nanoparticles are addressed to biomedical application, the study of magnetization and colloidal stability in physiological conditions should be tested. The presence of salts in solution and/or presence of proteins (i.e. BSA) could modify the magnetization if the colloidal stability is comprised or increased. Have you some information in this respect? What is the limit concentration to have a good colloidal suspension? The colloidal stability remains after magnetization?.
It is difficult to think that these nanoparticles are very stable in aqueous solution without the presence of a stabilizer as different biocompatible polymers (PEG, PLGA, Chitosan). Do the authors believe that if these types of stabilizers are used, they would radically change the magnetic susceptibility of these nanoparticles?. A discussion in this aspect would be recommendable is the final application will be on biomedical field.
Graphics should be homogenized in size. In some figures (i.e. figure 7, 9, 10) the labels, numbers, etc have different sizes.
Author Response

(The authors gave the same response as above.)
